# Coastal complexity of the Antarctic continent

Richard Porter-Smith[1], John McKinlay[2], Alexander D. Fraser[1,3] & Robert A. Massom[1, 2]

[1]Antarctic Climate & Ecosystems Corporative Research Centre, University of Tasmania, Hobart, Tasmania, Australia
[2]Australian Antarctic Division, Kingston, Tasmania, Australia
[3]Institute for Marine and Antarctic Studies, University of Tasmania, Hobart, Tasmania, Australia

*Correspondence to*: R. Porter-Smith (r.smith@utas.edu.au)

**Abstract.** The Antarctic outer coastal margin (i.e., the coastline itself, or the terminus/front of ice shelves, whichever is
adjacent to the ocean) is a key interface between the ice-sheet and terrestrial environments and the Southern Ocean. Its physical configuration (including both length scale of variation and orientation/aspect) has direct bearing on several closely associated cryospheric, biological, oceanographical and ecological processes, yet no study has quantified the coastal complexity or orientation of Antarctica's coastal margin. This first-of-a-kind characterisation of Antarctic coastal complexity aims to address this knowledge gap. We quantify and investigate the physical configuration and complexity of Antarctica's circumpolar outer
coastal margin using a novel technique based on ~40,000 random points selected along a vector coastline derived from the MODIS Mosaic of Antarctica dataset. At each point, a complexity metric is calculated at length scales from 1 to 256 km, giving a multiscale estimate of the magnitude and direction of undulation or complexity at each point location along the entire coastline. Using a cluster analysis to determine characteristic complexity 'signatures' for random nodes, the coastline is found to comprise three basic groups or classes: (i) low complexity at all scales; (ii) most complexity at shorter scales; and (iii) most
complexity at longer scales. These classes are somewhat heterogeneously distributed throughout the continent. We also consider bays and peninsulas separately and characterise their multi-scale orientation. This unique dataset and its summary analysis have numerous applications for both geophysical and biological studies. All these data are referenced by doi:10.26179/5d1af0ba45c03 (Porter-Smith et al., 2019) and are available free of charge at http://data.antarctica.gov.au.

**Copyright statement**

……………………………………………

## 1    Introduction

Although substantial research has been undertaken on quantification of coastal complexity of terrestrial areas (Andrle, 1996a;
Bartley et al., 2001; Jiang and Plotnick, 1998; Porter-Smith and McKinlay, 2012), equivalent attention has not yet been paid to the polar continents and ice sheets i.e., the Antarctic and Greenland. Over its vast total length of >30,000 km (Figure 1), the coastline of Antarctica – the focus of this study - comprises only 5% exposed rock - the remainder consists of ice at the seaward margins of either: (i) ice sheet grounded (resting) on bedrock (38%); or (ii) floating extensions of the ice sheet in the form of ice shelves (44%) and glacier tongues or snouts (13%) up to several hundred metres thick (Drewry et al., 1982). As such, the
Antarctic coastline is more dynamic than its mid-latitude terrestrial counterparts due to ice advance and iceberg calving, and its complexity is therefore more challenging to quantify (Porter-Smith, 2003). Given its direct contact with the high-latitude ocean and atmosphere, the floating outer margin of the ice sheet is also highly sensitive to climate and environmental change.

Characterising the magnitude and direction of bays and peninsulas over a range of length scales and aspects is a necessary step to evaluating the important though poorly understood effects of coastal complexity on key physical, ecological and biological processes and phenomena occurring around the circumpolar Antarctic margin. Localised case studies have shown coastline geometry/aspect to be a major determinant of the distribution and properties of sea ice in the Antarctic coastal zone (Fraser et al., 2012; Giles et al., 2008; Massom et al., 2001), and to affect important ice sheet margin processes e.g., ice shelf-ocean interaction, melt and iceberg calving – with implications for sea-level rise.

Notably, coastal complexity (here termed $Cx$) is likely to be an important factor determining the observed variability in patterns of spatial extent and persistence of landfast sea ice (fast ice) around the Antarctic coastal zone (Fraser et al., 2012), where fast ice forms both dynamically through interception of pack ice by coastal protrusions (and grounded icebergs) and thermodynamically in sheltered embayments (Fraser et al., 2012; Giles et al., 2008; Massom et al., 2001). Developing improved knowledge of factors affecting fast ice distribution and polynya behaviour – including coastline configuration - is a high priority, as a first step towards better predicting the likely future trajectory of the vulnerable Antarctic coastal environment in a changing climate. Fast ice forms a crucially important habitat e.g., for Emperor penguins (Massom et al., 2009), is a determinant of ice-shelf stability (Massom et al., 2010; 2018), and has a major impact on logistical operations, e.g., station resupply.

Moreover, coastline configuration and fast-ice distribution are primary determinants of the location and size of Antarctic coastal polynyas (Massom et al., 1998; Fraser et al., in press). Antarctic polynyas are of regional to global significance as sites of high sea-ice production and (in certain cases) associated Antarctic Bottom Water (AABW) formation (Rintoul, 1985) that drives global ocean thermohaline circulation. Polynyas are also areas of enhanced biological productivity and form key habitat for marine mammals and birds (Arrigo and van Dijken, 2003; Tynan et al., 2010).

A further motivation relates to improving model simulation of the complex and highly vulnerable Antarctic coastal environment, and the processes therein. Although model representation of coastlines is inherently smoother than reality due to limitations of model resolution (Hibler, 1979), coastal complexity is a consideration for producing more accurate dynamic sea ice models, whereby "rougher" coasts (with higher $Cx$) tend to favour production of shear margins/zones in the mobile offshore pack ice. For sea ice models with insufficient spatial resolution to resolve $Cx$ explicitly, parameterisation of $Cx$ is required – but baseline knowledge of $Cx$ is currently lacking. Such a dataset can provide a "roughness" boundary for sea ice models which currently have insufficient spatial resolution to explicitly resolve the coastline. Characterisation of coastline complexity magnitude, feature type (embayment or peninsula) and feature aspect could also feed into exposure models for wave-ice shelf interaction (Manson et al., 2005; Massom et al., 2018) and studies quantifying wave exposure relative to coastline features. This would naturally complement general fetch and exposure models (Hill et al., 2010; Reid and Massom, 2016 updated 2019).

The complexity of terrestrial coastlines is dependent on geological inheritance and surrounding ocean processes. For instance, each of Australia's geological regions displays discrete complexity signatures demonstrating a correlation between coastal complexity and geology – an analysis of which revealed a close relation between $Cx$, lithological mix and ocean processes e.g., Porter-Smith and McKinlay (2012). These signatures can vary enormously, between regions and over a range of length scales. Geological phenomena cannot be captured by a single value (Ringrose, 1994), and an attempt to do so may cause a process or form to be missed or misinterpreted. To capture the true complexity of the coastline, it is necessary to adopt a method that accounts for scale variation, since geomorphological features and associated processes can vary across several scales (Andrle, 1994, 1996a; Goodchild and Mark, 1987; Lam and Quattrochi, 1992). Therefore, an appreciation of the variability of complexity evident at different length scales is crucial (Porter-Smith and McKinlay, 2012).

Characterisation of the complexity of terrestrial coastlines is a fundamental measure of the lithological mix. Coastlines of a homogeneous lithology tend to be straighter than coastlines of mixed lithology. Wave action promotes a straight coastline if the lithology is homogeneous and a complex one if the lithology is heterogeneous (Porter-Smith and McKinlay, 2012). The Antarctic coastline is a different challenge in that it is almost totally covered by glacier ice and surrounded by ice barriers that influence ocean processes acting on the continent and is likely to be more likely to be more temporally-variable in nature than terrestrial coastlines. Additionally, knowledge of the underlying rock type is severely limited due to inability to access much of the geology through the ice (Stål et al., 2019).

However, even in this homogeneous environment, one might expect a relatively higher complexity due to the presence of glacial valleys, an example would include the western Peninsula's fjord-like coast, where there are glacial erosion processes in motion. Glacial erosive processes have a distinct signature (Anderson et al., 2006) that would result in a higher coastal complexity. Although the formative processes may differ between Antarctic and terrestrial scenarios, the methodology does not assume prescriptive or formative processes but classifies purely on differences in complexity over a range of length scales. The analysis of *Cx* using this multi-scale approach also allows the identification and analysis of morphologically similar coastal environments and forms the basis for further research into their relationship to, and synergy with natural processes.

Despite its importance, no study has quantified the coastline complexity of the Antarctic continent. Here, we address this critical gap by carrying out a first quantification of the geometric configuration and complexity of the Antarctic coastline, using a novel technique to examine the spatial distribution of both the magnitude and direction (aspect) of *Cx* over varying length scales. This new dataset (Porter-Smith et al., 2019) not only highlights spatial differences but, serves as an important yardstick against which to gauge future change and variability in coastal complexity and character around Antarctica. In this study, we derive methods for determining scale-dependent metrics describing coastal complexity of the Antarctic continent, including the facility to classify points as belonging to bays or peninsulas at different scales. Using this metric at 40,000 random point locations around the coastal margin, we use clustering techniques to determine characteristic complexity "signatures" around the continent.

## 2    Methods

### 2.1    Quantifying the complexity of the Antarctic coastline

To calculate *Cx* for the entire Antarctic continental coastline, 40,000 points were randomly chosen along the MODIS Mosaic of Antarctica 2008-2009 (MOA2009) coastline dataset (Haran et al., 2014), acquired from the US National Snow and Ice Data Center (NSIDC). The coastal margin is used in the calculation of complexity since the outer margin is more relevant for the processes listed in the introduction here (e.g., ecological habitats, fast ice formation, polynya location, ice shelf-ocean interaction). Figure 2 illustrates the algorithm for determining *Cx*. At each random target point *x* on the merged MOA dataset, and for each length scale (of 1, 2, 4, 8, 16, 32, 64, 128 and 256 km), the Euclidean straight-line distance was measured either side of the chosen point to find the corresponding points, *a* and *b*, that intersect the coastline. The two vectors $\overline{xa}$ and $\overline{xb}$ are vector-summed to give the quantity $\overline{xc}$, indicating the magnitude of complexity, and direction (both relative to north and the local coastline) for the aspect. The maximum distance between successive random points was rarely greater than 1 km, thereby giving a near-uniform and seamless representation of complexity around the continent (Figure 2).

This approach varies from previous techniques employed to derive *Cx*, such as the angled measurement technique (AMT) where the length scale is measured forward and backwards of a chosen point on the mapped coastline. In the AMT, the measure of complexity is the supplementary angle (Andrle, 1994, 1996b; Porter-Smith and McKinlay, 2012). The new approach

presented here; offers not only a measure of complexity (as magnitude) but also direction. Additionally, the new technique allowed qualification of the chosen section of coastline as either a bay or peninsula for a given length scale (i.e., any angle less than 180° would be classed as a bay, and any angle over 180° would be classed as a peninsula). An advantage of our technique is that it can be used to quantify coastal complexity at various scales to reflect the multi-scale nature of features along the

120    coastline. Additionally, characterising the orientation (i.e., aspect) of features is useful in that it can be compared to the directions of other potential co-variates, allowing correlations and interactions to be examined.

Given that complexity magnitude varies as length scale changes, the resultant magnitudes of $\overline{xc}$ were normalised to a range of 0 to 100 to give comparability between length scales. The spectrum of length scales examined was chosen to provide complexity measurements at scales relevant to known oceanic, cryospheric and geomorphological processes and phenomena

125    at kilometre-to-mesoscale levels, with individual lengths chosen as a series of base 2 powers to minimise the potential problem of spatial autocorrelation (Goodchild, 1986).

Data processing, spatial analysis and mapping was carried out using the GIS and spatial analysis platforms Arc/Info (ESRI, 1996) and QGIS (Quantum GIS Development Team, 2014). Statistical analysis was carried out using the R language for statistical computing (Ihaka and Gentleman, 1996; R Core Team, 2014) and the R package *cluster* (Maechler et al., 2018).

130    **Table 1: List of fields and their descriptions. (*Repeated for length scales 1, 2, 4, 8, 16, 32, 64, 128 and 256 kms for each point).**

| Variables | Definitions |
|---|---|
| STATION | Station number |
| EASTING | Easting Polar Stereographic |
| NORTHING | Northing Polar Stereographic |
| X_COORD | X geographic coordinate |
| Y_COORD | Y geographic coordinate |
| COAST_EDGE | Type of coast 'Ice shelf/Ground' |
| *FEAT_01KM – 256KM | Described feature 'Bay/Peninsula' |
| *AMT_01KM – 256KM | Measure of complexity, Angled Measurement Technique 0-180 degrees |
| *MAG_01KM – 256KM | Measure of complexity - Magnitude on dimensionless scale 0-100 |
| *ANG_01KM – 256KM | Angle (absolute angle of station points from reference 0, 0) |
| *ANGR_01KM – 256KM | Angle of 'Magnitude' (relative to coastline - directly offshore being 0/360°) |

## 2.2    Clustering

Unsupervised classification (clustering) techniques were used to determine how many distinct complexity classes exist around

135    the Antarctic coast. Cluster analysis has a rich history in statistics and machine learning (Hastie et al., 2001; Kaufman and Rousseeuw, 1990). In both fields, it is primarily used as an exploratory technique to identify $k$ groups from $n$ observations, such that observations within groups are more similar to one another in their $p$ multivariate responses than they are compared with those in other groups.

Given the large size of the dataset, and the high computational burden of many clustering algorithms, two common and

140    tractable methodologies were selected: $k$-means and partitioning around medoids ($k$-medoids) (Kaufman and Rousseeuw, 1990; Maechler et al., 2018). These centroid-based partitioning methods were applied to the $n \approx 40,000$ complexity magnitude values for $p = 9$ length scales (i.e. 1, 2, 4, 8, 16, 32, 64, 128 and 256 km). For both $k$-means and $k$-medoids, length scales were first standardised (0-4116100) and Euclidean distances were used as the metric describing the similarity between observations. The primary difference between these clustering techniques is that while $k$-means attempts to group objects into $k$ clusters

based on minimising the distance of observations to group means (i.e. minimising the within-cluster sums-of-squares), $k$-medoids operates by minimising distances to group medoids, where the latter are data points that are analogous to multivariate medians. Thus, clustering by $k$-medoids can be considered a robust alternative to $k$-means that will be less influenced by outliers and noise in the data. Given the size of the merged MOA coastline dataset, we employ the Clustering LARge Applications (CLARA) implementation of partitioning around medoids, a method that subsets data in order to achieve an optimal solution that is linear, rather than quadratic, in $n$. The algorithm of Hartigan and Wong (1979) was used for $k$-means clustering, and optimisation was conducted over several random starts to ensure global optimisation was achieved.

For any given application, clustering should be carried out for the spatial extent and at spatial scales relevant to the phenomena under investigation. As the present study seeks a synoptic, Antarctic-wide summary of complexity, we first consider all data (Antarctic-wide, all length scales) in a single analysis. In this case, all length scales are afforded equal weight in the analysis. However, it is likely that many local- to regional-scale phenomena impacting oceanic and cryosphere processes may be relatively unaffected by smaller scale complexity. For this reason, cluster analyses were repeated on complexity data restricted to length scales >=8 km and results compared with those derived from analyses of all length scales considered simultaneously.

### 2.3    Gap statistic for determining number of clusters

A common problem when conducting unsupervised classification is that often the true number of groups, say $k^*$, is unknown and must be estimated from the data. Estimating $k^*$ is a difficult and somewhat ill-defined problem since there is no universal definition of what should constitute a group, and this has led to a wide variety of approaches for estimating $k^*$ under different clustering scenarios (Charrad et al., 2014; Milligan and Cooper, 1985). The Gap statistic, which can be used in conjunction with many clustering techniques, is one of the more useful approaches to objectively determining $k^*$ (Tibshirani et al., 2001). While it is known to perform imperfectly in a limited set of circumstances (Mohajer et al., 2011), Tibshirani et al (2001) use simulation experiments and analyses of real data to demonstrate that the technique outperforms a wide range of alternate established methods. The technique determines the optimal number of groups by examining the within cluster dispersion $W_k$ as a function of the number of clusters $k$. Obtaining separate clustering solutions for $k \in \{1, 2, \ldots, k_{max}\}$, along with corresponding $W_k$ values $\{W_1, W_2, \ldots, W_{k_{max}}\}$, shows that by itself $W_k$ is uninformative since it always decreases with increasing $k$, even for independent data with no structure. The Gap statistic overcomes this problem by defining

$$\mathrm{Gap}_n(k) = E_n\{\log(W_k)\} - \log(W_k) \qquad \forall k, \, k = 1, 2, \ldots, k_{max}$$

where $E_n$ denotes the expectation under a sample size of $n$ from a reference ($H_0$) distribution. The latter is determined by resampling from a uniform distribution on the $p$-hypercube determined by the ranges of the data after first centring and rotating them to align with their principal axes. The optimal cluster number $k^*$ is estimated as the value maximising $\mathrm{Gap}_n(k)$ after considering sampling variability associated with determining the reference distribution. In practice, this is achieved by choosing $k^*$ to provide the maximum Gap statistic that is within 1-standard-error (Breiman et al., 1984) of the first local maximum over the range of $k$ (Tibshirani et al., 2001). For the present study, $E_n\{\log(W_k)\}$ was estimated by an average of 100 separate Monte Carlo samples of the reference distribution. For both $k$-means and $k$-medoids, $\mathrm{Gap}_n(k)$ was assessed over the range k=1 to 20. The gap statistic can be calculated for a range of clustering algorithms, which allows the similarity in clustering solutions to be compared between methods.

# 3    Summary

## 3.1    Complexity and aspect around the continent

The total length of the outer merged MOA coastline is 39,593 km. The length of ice shelf and grounded ice coastline around the continent are 21,269 km and 18,324 km respectively and roughly proportional in western and eastern Antarctica. There is a strong positive-skew in the distribution of $Cx$ at all length scales, and this skew is especially pronounced at shorter length scales, i.e., complexity is not normally-distributed, indicating that the Antarctic coastal margin has a tendency to be straighter rather than highly complex (Figure 3).

A notable difference between western and eastern sectors of Antarctica (-180-0° and 0–180°) is the orientation of both bays and peninsulas. In East Antarctica, these features generally face directly offshore across all length scales (Figure 4), with the higher $Cx$ magnitude generally facing directly offshore, i.e., a normal distribution of magnitudes and their orientations. In West Antarctica, on the other hand, both bays and peninsulas have a general skew toward the west-of-offshore direction. This becomes particularly dominant at length scales of >16 km. This bias in the bay and peninsula feature orientation may have implications for key physical processes (e.g., formation and persistence of fast ice) and biological processes highlighted in Section 1.1. These variances could be used to examine and differentiate between regional and local areas, and with other co-variates to analyse specific phenomena.

## 3.2    Determining the number of complexity groups using clustering

Analysis of the gap statistics shows that omission of smaller length scales (<= 8km) produces a pronounced local maximum at k=3. This suggests that the optimal number of complexity groups is three, as shown by the 'elbow' in the gap statistic plots (see Figure 5).

**A projection of random point scores onto the first two principal component axes, accounting for 41% of the total variation (**

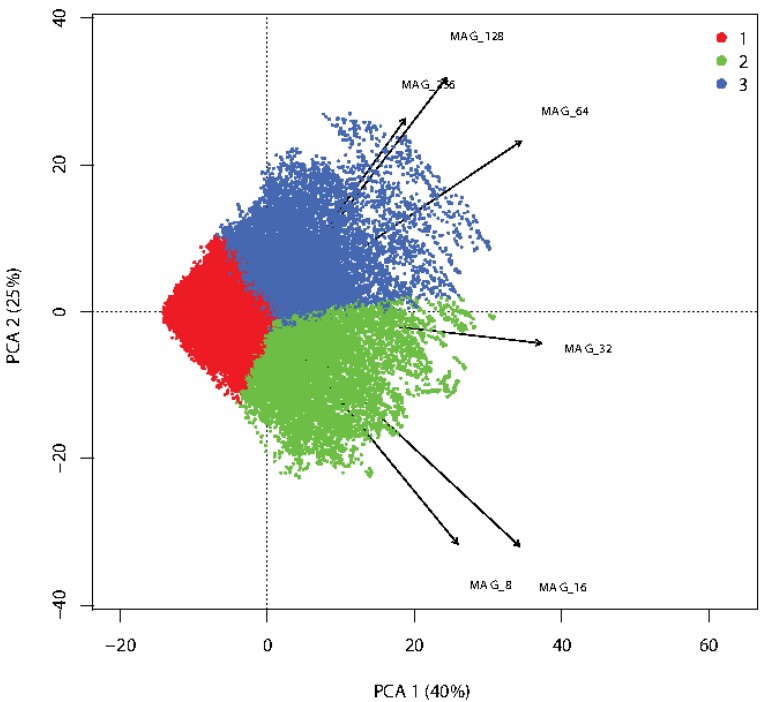

Figure 6), shows the three groups in relation to projections of the complexity length classes. As might be expected, arrows representing the complexity length classes appear in approximate order, in a fan shape, indicating that adjacent classes are most closely correlated with one another. Variances look approximately the same (i.e., arrows are approximately the same length) across length classes. In the two-dimensional approximation, the three groups show considerable overlap.

Figure 7 shows violin plots of $Cx$ magnitude (for $>= 8$ km length scales), by the three-group structure determined by PAM clustering. The red line joins adjacent medians. This plot reveals the multiscale complexity of each group: Group 1 represents coastline with little complexity (i.e., relatively smooth) at all length scales; Group 2 represents coastline with more small-scale ($<= 32$ km) complexity; and Group 3 represents coastline with more large-scale ($>= 64$ km) complexity.

The $Cx$ dataset (Porter-Smith et al., 2019) presented here allows spatially resolved characterisation of normalised complexity as a function of longitude, for each length scale. This is shown in Figure 8 as a polar plot. For simplicity, we show only the normalised complexity for 16 km (representing the "class 2" short length-scale cluster) and 128 km (representing the "class 3" long length-scale cluster). For both bays and peninsulas, the 16 km $Cx$ is both larger and more homogenous as a function as longitude (bays: mean $Cx$=24.8 +/- 16.7; peninsulas: mean $Cx$= 23.9 +/- 15.0), whereas the 128 km $Cx$ is more heterogeneous/episodic in nature (bays: mean $Cx$=13.0 +-/ 17.3; peninsulas: mean $Cx$=15.1 +/- 17.2).

Figure 9 shows the mix of coastline groups contained within a 64 km sliding window (chosen to allow as many data points as possible while still representing reasonably short length-scale variability) for the entire coastal margin. Although groups or typologies are observed to occur heterogeneously around the entire coastline, certain classes tend to dominate at specific scales and locations around the continent. To derive the dominant group within the heterogeneity, each of the three groups were totalled within the sliding window and proportionately normalised to 255. The dominance and heterogeneity could then be expressed and represented as a value within the RGB colour model.

As expected, the coastal margins of the Ronne, Ross and Larsen C ice shelves are predominantly Group 1. This reflects the very smooth nature of these ice shelf fronts, which tend to calve large, tabular icebergs. There are also several other ice shelves exhibiting Group 1 dominance but which do not calve large tabular icebergs, including the Larsen D ice shelf on the eastern side of the Peninsula, the Venable and Abbots ice shelves on the western side of the Peninsula, and the ice shelves of the Sabrina Coast of East Antarctica. Several East Antarctic regions of grounded ice margin also exhibit Group 1 dominance, including the Prince Olav, Mawson, Ingrid Christensen, Wilhelm II, Knox, and Wilkes and Adélie Land coasts.

Regions dominated by Group 2 (indicating high $Cx$ at small length scales) include the grounded ice coastal margin on the northern part of the western Antarctic Peninsula (between Cape Roquemaurel at 63.5° S, 58.9W and Cape Jeremy at 69.4°S, 68.8W); a mountainous stretch of Victoria Land on the coast of the Western Ross Sea which is punctuated by glacier tongues of length 15 to 25 km (between Cape Washington at 74.7°S, 165.5°E and Coulman Island at 73.3°S, 169.7°E); and the Sulzberger Ice Shelf region (at 77°S, 150°W). The latter is characterised by a highly crevassed and rough (on a 25 km scale) ice shelf margin resulting from severe dynamical constraints on outflowing glacial ice.

Regions exhibiting Group 3 dominance, on the other hand, occur mainly at major coastal inflection points. Notable locations are where the Transantarctic Mountains meet the McMurdo Ice Shelf; at the tip of the Antarctic Peninsula); and along the coastline of Alexander Island and the Wilkins Ice Shelf. There, coastal undulations occur on the large spatial scale captured by Group 3 (64 to 256 km).

Enlargements of Figure 9 around Enderby Land and Victoria Land are presented in Figure 10 and Figure 11, respectively. These enlargements highlight regions of complex heterogeneity in Cx.

## 4     Data availability

These data are available free of charge from the Australian Antarctic Data Centre ([http://data.antarctica.gov.au](http://data.antarctica.gov.au)) and are referenced by [doi:10.26179/5d1af0ba45c03](doi:10.26179/5d1af0ba45c03) (Porter-Smith et al., 2019).

## 5     Conclusions

This first-of-a-kind study of Antarctic coastal complexity has quantified and classified discreet morphology signals using a novel technique to produce a new dataset describing complexity for the entire circum-Antarctic coastal margin over a range of scales. To date, there has been no quantification of the physical configuration of this important interface, despite its central relevance to other research areas. Here, we show that the Antarctic coastal margin is generally straighter than the coastlines of typical terrestrial continents; this is likely due to the generally uniform mechanical strength of the ice compared to the mixed lithology and resultant higher complexity promoted by erosive processes of terrestrial landforms. Another key finding is that, based on the multi-scale complexity characterisation, the Antarctic coastal margin can be classified into three main groups; these are (i) low complexity; (ii) complex at short length scales, and complex at long scales). While the Antarctic coastline is largely found to be spatially heterogeneous in its physical complexity, there are dominant groups along certain individual stretches. This study has also, for the first time, quantified and characterised specific Antarctic coastal features such as bays and peninsulas and their orientation at various length scales. Another key finding is that the aspect (orientation) of bay and peninsula features is different for western and eastern Antarctica.

Given the temporally variable nature of ice and as to the question of how frequently the complexity of the Antarctic coastline should be recalculated, most major change in margins happens with ice shelf advance/retreat (i.e., calving and ice front advance). Of these processes, retreat has by far a shorter timescale. So, one could argue that a re-assessment should happen in conjunction with major calving - but such events tend to be regionally limited (e.g., the calving of the Amery Ice Shelf in 2020). Ice shelf collapse (e.g., Wilkins in 2008/09) is a little more dramatic but still geographically limited. Thereby, such re-evaluations aren't needed frequently unless there's major change. Runaway grounding line retreat leading to major coastal margins changes might be sufficient grounds for re-evaluation, but this hasn't yet happened. Significance of changes could be assess using standard change detection metrics (e.g., estimate the distribution of the current coastline features, see if the new coastline complexity falls outside of this distribution) thus justifying another evaluation.

Our complexity definition methodology provides a quantitative, repeatable approach to analysing coastline features, and could be readily applied to other coastlines both in terrestrial and polar regions. This unique dataset and its analysis presented here also have numerous applications for both geophysical and biological studies and will contribute to Antarctic research requiring quantitative information on (and related to) coastal complexity and configuration. For instance, and in the crucially important field of modelling, a measure of coastal complexity provides a "roughness" boundary, thereby providing a parameterisation that is currently missing e.g., towards more accurate dynamic sea ice models. Similarly, and for general ocean fetch (wave) models, the characterisation of coastline complexity magnitude, feature type (embayment or promontory) and their aspect could also feed into exposure models for study of wave-ice shelf interaction, wave exposure and high-low energy habitat types.

**Author contribution**

RPS designed the methodology, compilation of data and analysis with contributions from all co-authors. JM provided statistical
guidance. RPS prepared the manuscript with contributions from all co-authors."

**Competing interests**

The authors declare that they have no conflict of interest.

**Acknowledgements**

We would like to thank two reviewers for their highly constructive and insightful comments on an earlier version of this manuscript. We would like to acknowledge the National Snow and Ice Data Center (NSIDC) for their provision of MODIS Mosaic of Antarctica coastline dataset. This work is supported by of the Australian Government's Cooperative Research Centre program through the Antarctic Climate & Ecosystems Cooperative Research Centre, and by the Australian Research Council's
Special Research Initiative for Antarctic Gateway Partnership (Project ID SR140300001). This work also contributes to the Australian Antarctic Partnership Program (AAPP) funded by the Australian Government Department of Industry, Science, Energy and Resources through the Antarctic Science Collaboration Initiative.

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

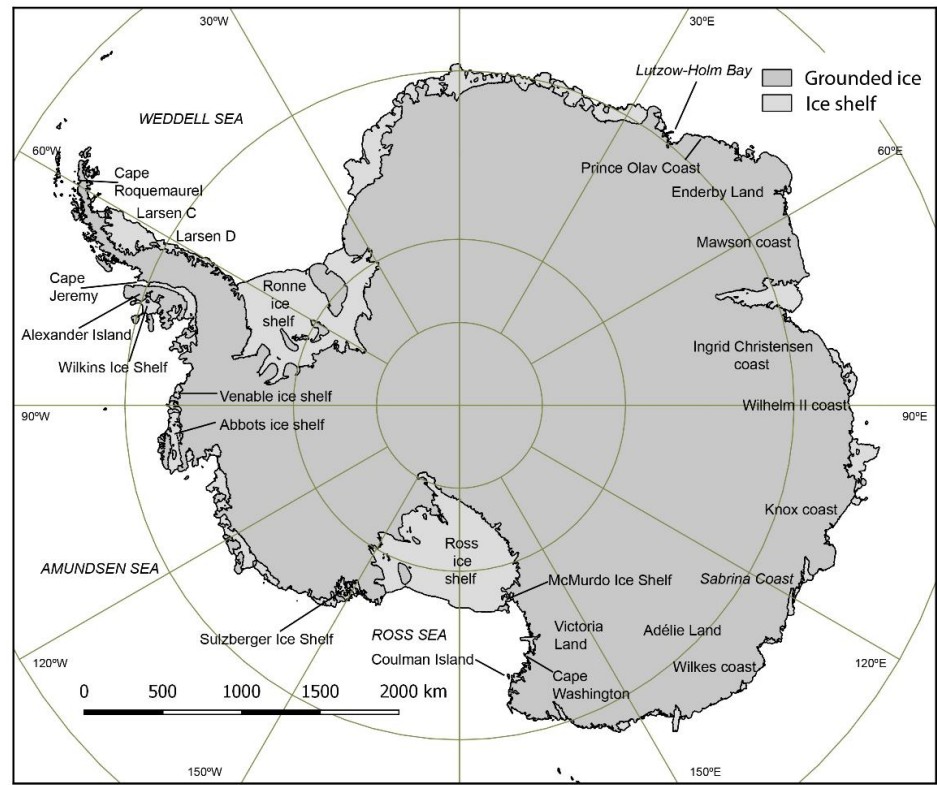


**Figure 1:  Map of the vector coastline derived from the MODIS Mosaic of Antarctica dataset (Scambos et al., 2007) showing the coastline, the distribution of ice shelves and (inland) the grounded ice coastline. Offshore islands were excluded for this study. Inclusion of islands is complex and application-specific, so will be considered in future detailed case studies of specific areas. Reproduced with permission.**


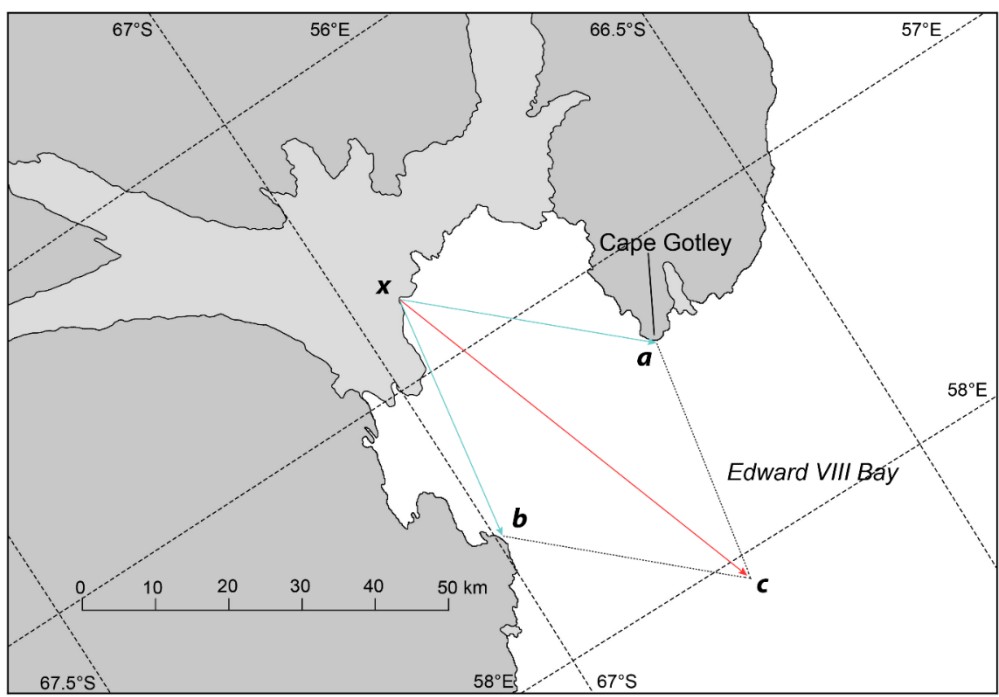

**Figure 2:  Example illustrating the calculation of complexity along part of coastline at Edward VIII Bay  at a length scale of 32 km using a vector-addition methodology. On the mapped-coastline, length scales  are measured either side of a point ($x$) intersecting the coastline at points $a$ and $b$. By adding these vectors, a measure of complexity is derived giving both magnitude and direction. In**
**addition, the angle between $\overline{xa}$ and $\overline{xb}$ provides the classification for bay/peninsula.**

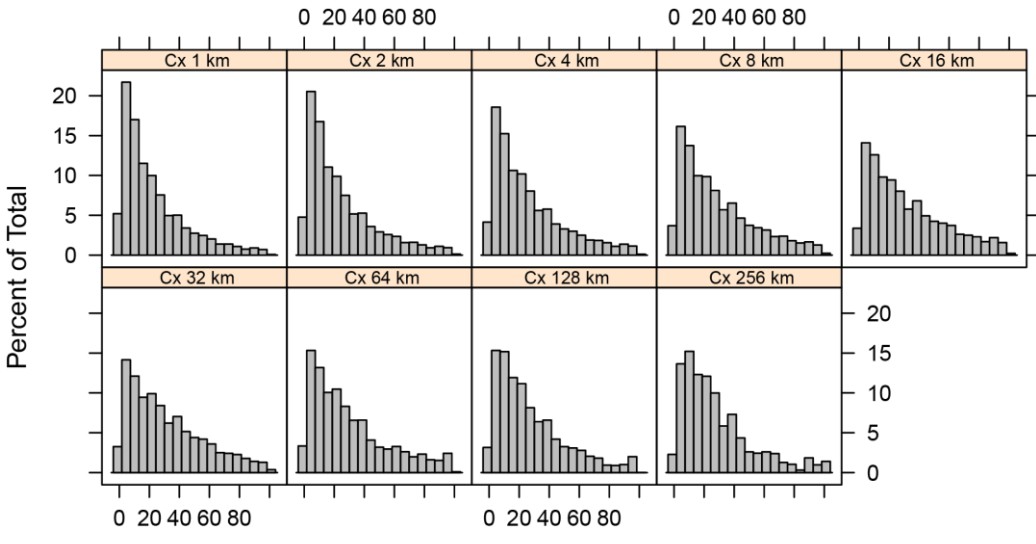

**Figure 3:** **Histograms of each MAG variable at each length-scale, showing a strong right skew for every length-scale**


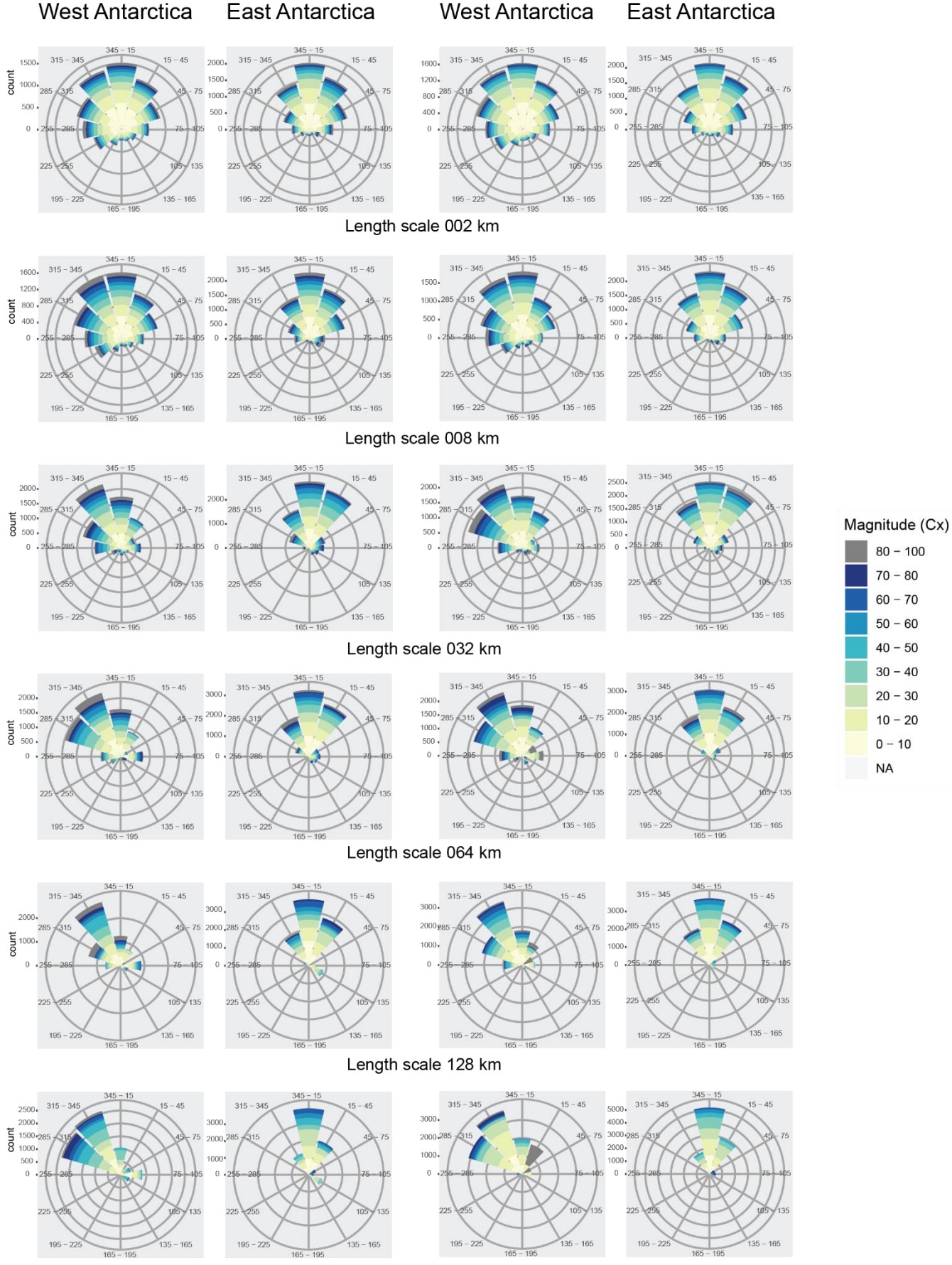

**Figure 4:** Rose plots comparing the distribution of complexity (magnitude and direction) between western and eastern Antarctic. Six length scales 2, 8, 32 64, 128 and 256 km are represented. The directions are relative to offshore.

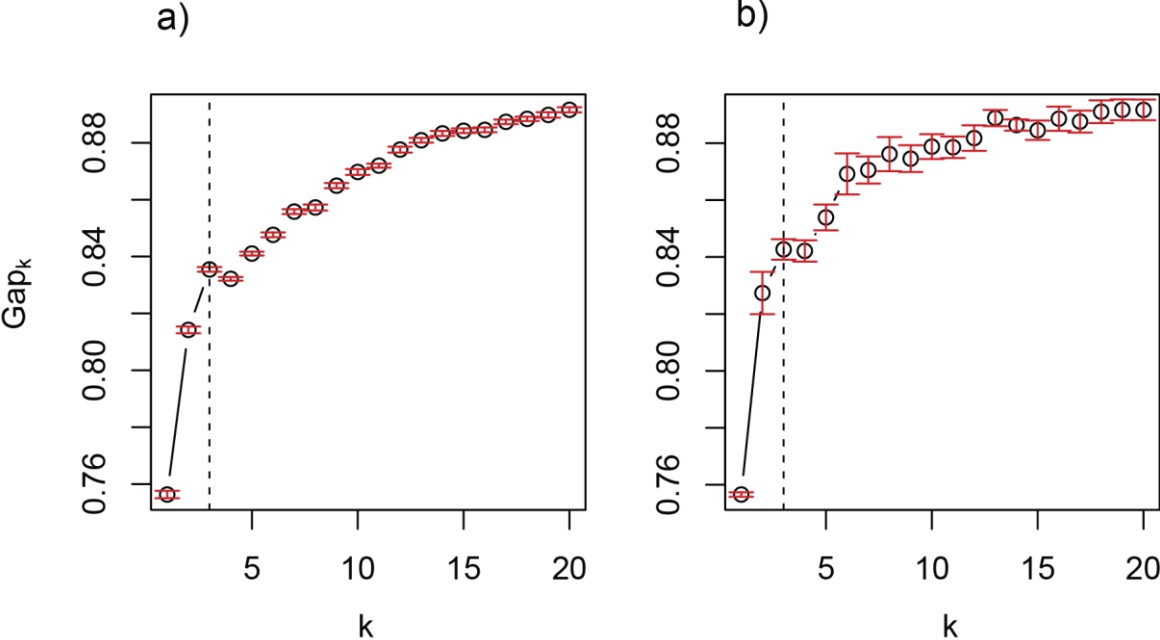

**Figure 5:** Gap statistic for length scales >=8 km produces a pronounced local maximum at k=3 indicated by 'elbow' in both *k*-means (a) and *k*-medoids (b) clustering.

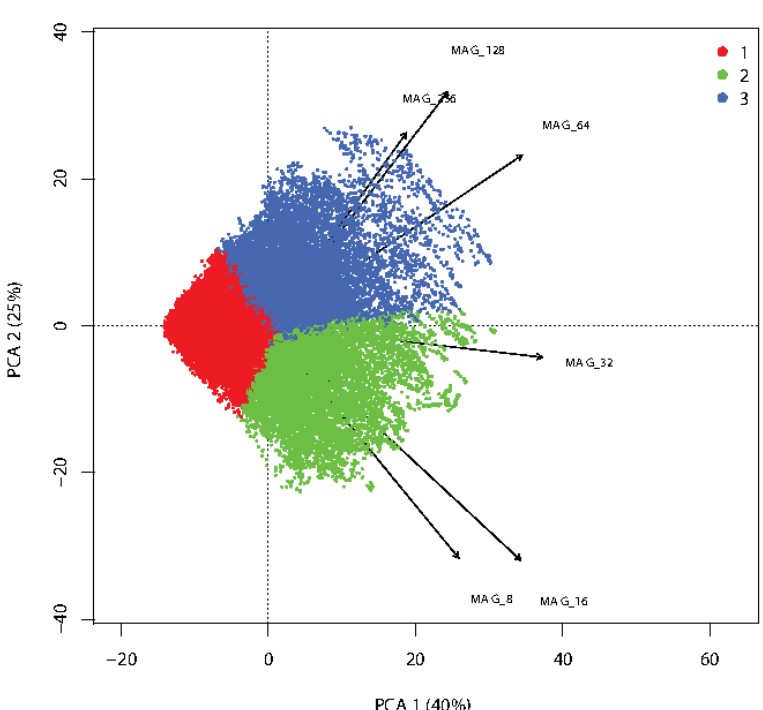

**Figure 6:** Principal components analysis biplot of *Cx* magnitude of complexity, **>= 8 km,** comparing the 3-Group structure determined using both k-means and PAM clustering.

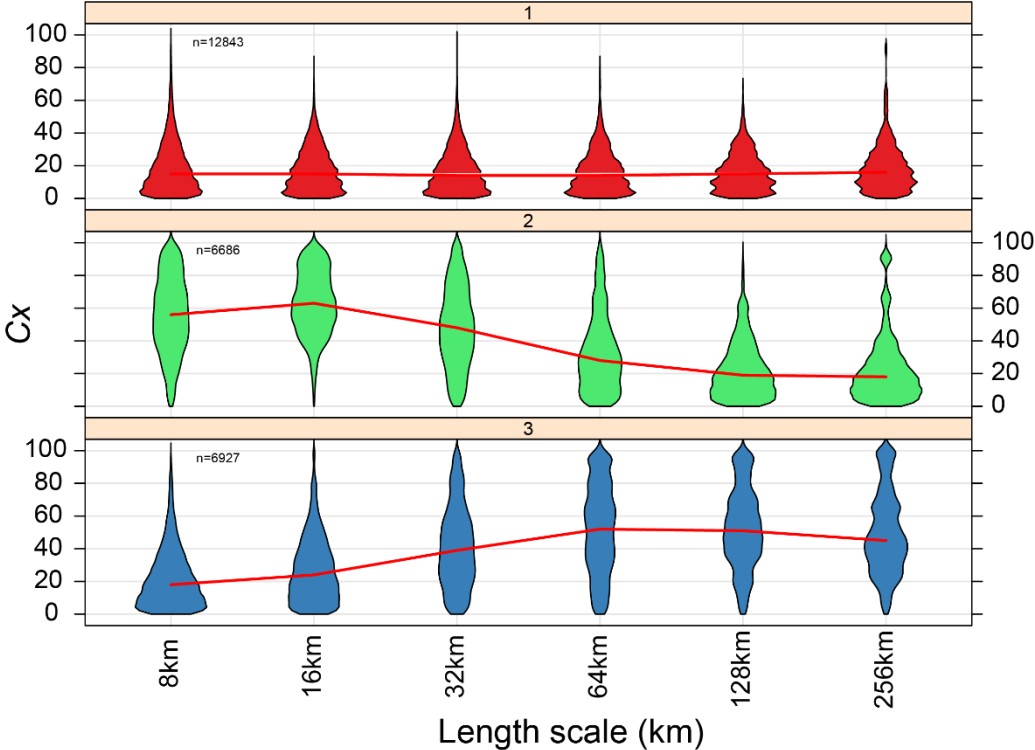

**Figure 7:** Violin plots of *Cx* magnitude values for >= 8 km length scales, showing 3-Group structure determined by *k*-medoids clustering. The red line joins adjacent median values in each distribution. Panel numbers indicate group number and are comparable with group numbers in allied plots.

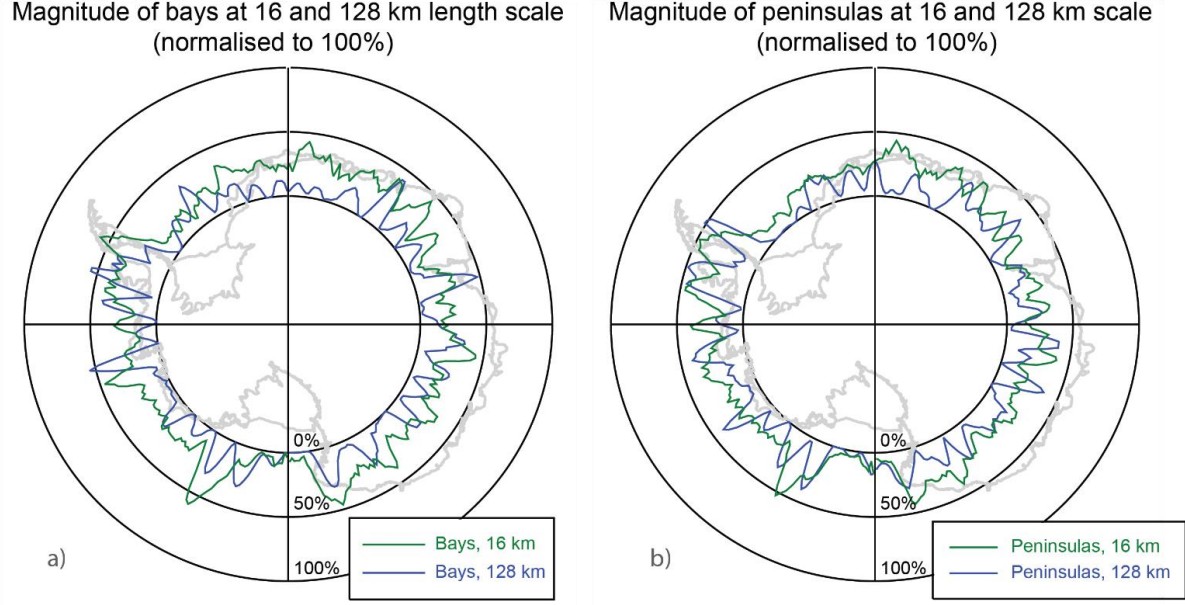


**Figure 8:** Polar plot showing normalised *Cx* for a) bays and b) peninsulas around Antarctica at various 16 km (i.e., "short", corresponding to group 2) and 128 km ("long"; group 3) scales, as a function of longitude. The coastline is shown in light grey.


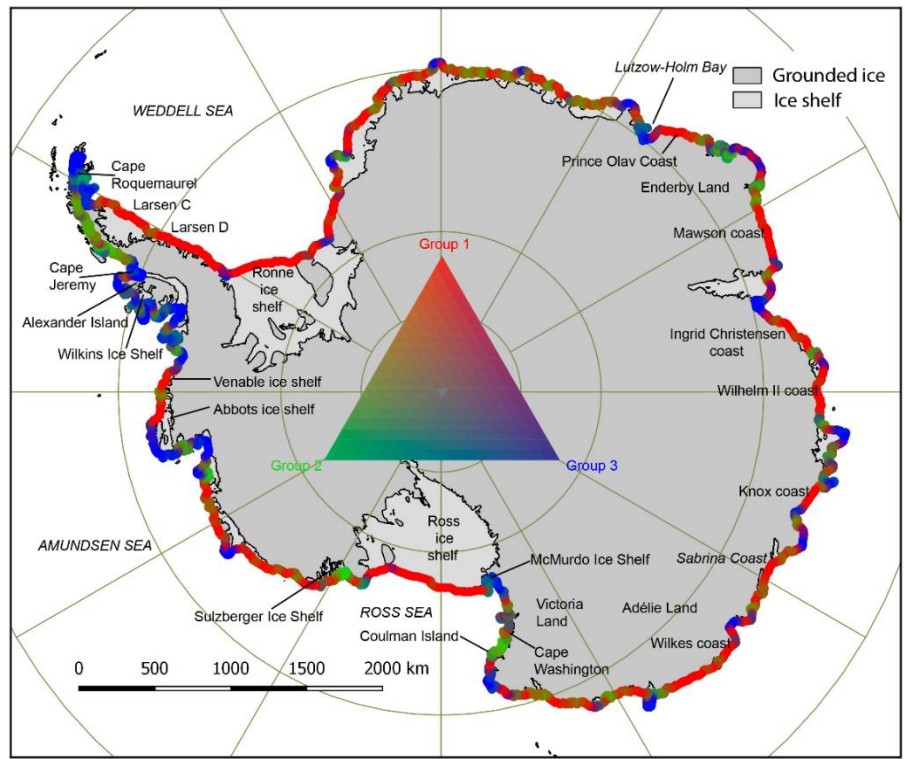

**Figure 9:   Spatial plot of 3-Group structure determined by PAM clustering, >= 8 km length scales.**

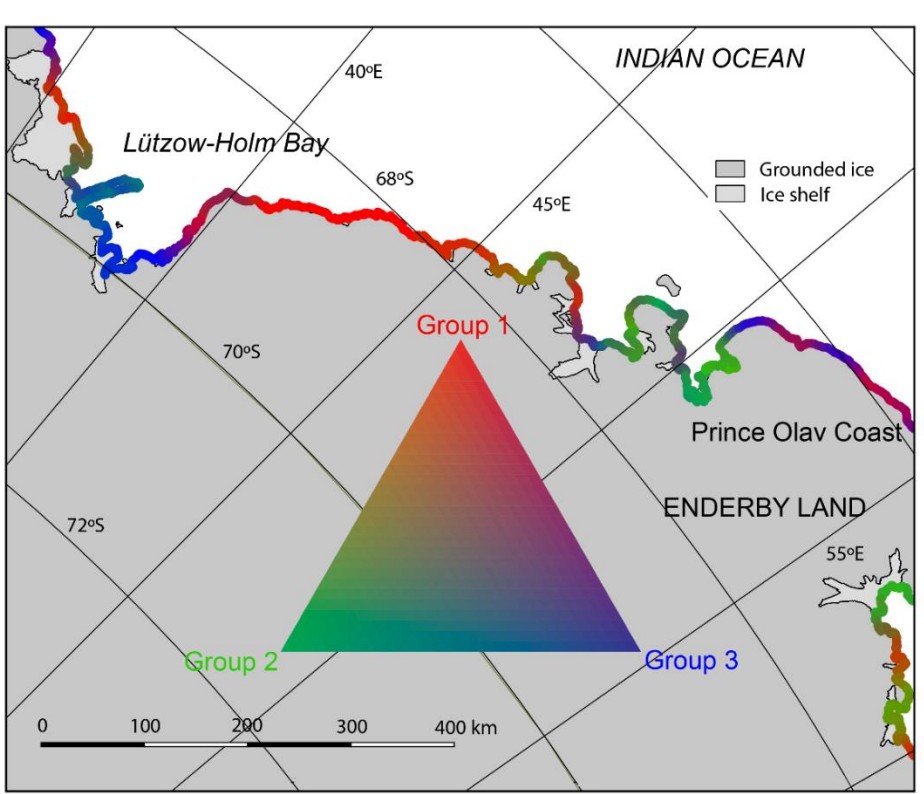

**Figure 10:   Enlargement of the Lützow-Holm Bay-Enderby Land-Prince Olav Coast region, showing a diversity of complexity classes. The deep (100 km across) Lützow-Holm Bay embayment is predominantly Group 3, whereas the remainder of the coast is either largely Group 1 (smooth stretch of Prince Olav Coast) or a mixture including Group 2 (north-eastern Enderby Land).**

**Lützow-Holm Bay is so deeply embayed that it shelters and favours formation of multiyear fast ice, whereas along Enderby Land coast, its much higher degree of exposure permits fast ice to form only seasonally.**


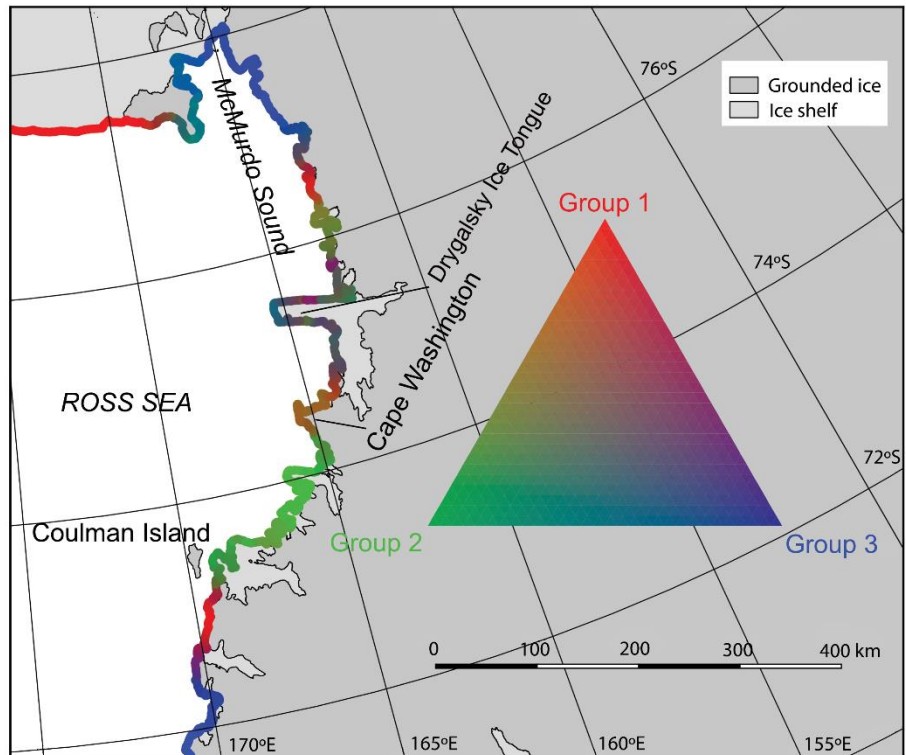

**Figure 11:   Enlargement of the western Ross Sea coastal area highlighting the diverse range of coastal classes. The smooth Ross Ice Shelf is predominantly Group 1, with Group 3 dominating the coastal inflection points at McMurdo Sound and Cape Adare. The remainder of the coast is largely mixed, except for a region of Group 2 from Cape Washington at 74.7° S to Coulman Island at 73.3° S. Fast ice forms readily within the protection of McMurdo Sound, but with incrementally lower persistence further north (Fraser et al., 2020). The northern and southern side of the Drygalsky Ice Tongue have identical complexity classes but very different glaciology (sometimes fast ice to the south, but almost always ice-free/polynya to the north. This highlights the role of embayment aspect, which is independent of complexity.**

