# Peer review of "Coastal complexity of the Antarctic continent"

_Earth System Science Data, 2019_

## Referee Comment (RC1) · Anonymous Referee #1 · 22 Sep 2020

This is a novel first attempt to characterize the coast of Antarctica, using techniques normally used for non-icy coasts. The significance of this technique is that it will allow the changes in coastal margins as the glacier/ice shelves retreat/change to be monitored. Problems may be that it is not clear how the "complexity" of an icy coastal margin relates to glacial processes. In the 'normal' coast, the recognition of bays etc is important for coastal erosion and deposition but within this icy sphere the link is not so obvious. So leading on from this, how frequently should the resurvey take place (daily, seasonal, annual decadal?), and how would you test the significance of the changes? Overall, I think this is a significant increase in knowledge about Antarctica and so should be published. I suggest the authors add more detail on – how the complexity adds to glacial processes, and plans for future resurvey.

---

## Referee Comment (RC2) · Ted Scambos (Referee) · 19 Mar 2021

The authors have produced an interesting and first-ever (for Antarctica) analysis of coastal complexity using methods previously used on other continents (Australia, at least). They describe the mathematics and statistics of the approach quite well (it seems clearly written and reasoned, and the math is not opaque), and note the -potential- importance and usefulness of the analysis in terms of studies of coastal ice and ocean conditions, and biodiversity.

The study is well-written and I have mostly minor comments. I would encourage the authors (and editor) to consider asking for some kind of 'case study' or close-up assessment of a region of interest, perhaps near Davis Base or McMurdo (places with lots of existing literature) and describe in qualitative or semi-quantitative terms how
the coastal type is reflected in the physical characteristics of the sea ice, biological environment and variability, or geology and erosion history..... it would make a stronger case that this is not just a topological exercise, but something that could potentially be mined for scientific application. The authors do a fair job of making this case in general terms, but two or three paragraphs looking at the results in a particular area woud be a good addition. Not an essential addition for publication, but one that would be nice to see.

Please also note the supplement to this comment:
https://essd.copernicus.org/preprints/essd-2019-142/essd-2019-142-RC2-supplement.pdf

―――――――――――――――――――

**Supplement:**

Review – ESSD, Coastal Complexity of Antarctica
    R. Porter-Smith et al.

L15 no comma needed between 'novel technique'
L22 suggest ending the sentence at '…biological studies'.  The last part has a circular reasoning tone to it.
L30 change to '…of terrestrial land areas.'
L37 is it really more challenging to quantify? Is it not just a mathematical analysis no matter the shape, or pace of change? Is change in the coastline over time considered somehow?
L40 I don't think the case for the importance of coastal complexity has really be made at this point in the text. Suggest moving this sentence to lead-off the paragraph that begins on Line 70.
L54 suggest '….and has a major impact on logistical operations, e.g., station resupply.'
L79 I think the text flow would be better if you moved this paragraph up one step and then combined the one above it and the one below it as the last statement before launching into the study.

L183-187  An interesting observation. I think the western-bias in West Antarctica may be an indirect function of climate and winds: much of the 0-60°W coastline (in your broad use of the word) is an ice shelf margin; but in the 60 – 180°W region, warmer conditions along the western Peninsula mean a more fjord-like coast, greater coastal complexity, and a strong westward orientation of both bays and peninsulas. On the Amundsen coast, there is a strong easterly prevailing wind direction, which may influence calving and ice shelf structure.   It would be a nice connection to the physical world to discuss the broad trends in this way (I'm not saying that my speculation is correct, but I think there is a physical / climate link to these obs of East and West Antarctica's coast).

L218 – change 'carve' to 'calve'

L233-234 Would it be useful / illuminating to include a third enlargement, perhaps of some part of the Peninsula? (I'm thinking southwest area or northern half on both sides).

Figure 6: I can't detect a difference between the left and the right diagram – ah, ok a very slight change in the red-blue boundary area…. But is this worth a separate figure? problem there?

Figure 11: change to 'Enlargement of the western Ross Sea coastal area…'

---

## Author Comment (AC1) · 12 Apr 2021

General comments

Anonymous Reviewer #1 ()

Comments: 'This is a novel first attempt to characterize the coast of Antarctica, using techniques normally used for non-icy coasts. The significance of this technique is that it will allow the changes in coastal margins as the glacier/ice shelves retreat/change to be monitored. Problems may be that it is not clear how the "complexity" of an icy coastal margin relates to glacial processes. In the 'normal' coast, the recognition of bays etc is important for coastal erosion and deposition but within this icy sphere the link is not so obvious. So, leading on from this, how frequently should the resurvey take place (daily, seasonal, annual decadal?), and how would you test the significance of the

changes? Overall, I think this is a significant increase in knowledge about Antarctica and so should be published. I suggest the authors add more detail on – how the complexity adds to glacial processes and plans for future resurvey.'

Response: To the reviewer#1, we thank you for taking the time to read and review our manuscript. Your insightful comments and suggestions – in particular, your reference to how frequently should re-evaluation take place given the temporally variability of the landscape – have gone a long way in helping us improve it to a way better standard. We have revised our original submission with your input and will submit the revised manuscript, along with a version containing all the changes made. Many thanks again.

We acknowledge that that we have generally talked about the influence of the complexity on other processes. As to the underlying reasons behind the changes in complexity, knowledge of the underlying rock type is severely limited due to inability to access much of the geology through the ice (Stål et al., 2019).

We clarified this point with modified text, "Characterisation of the complexity of terrestrial coastlines is a fundamental measure of the lithological mix. Coastlines of a homogeneous lithology tend to be straighter than coastlines of mixed lithology. Wave action promotes a straight coastline if the lithology is homogeneous and a complex one if the lithology is heterogeneous (Porter-Smith and McKinlay, 2012). The Antarctic coastline is a different challenge in that it is almost totally covered by glacier ice and surrounded by ice barriers that influence ocean processes acting on the continent and is likely to be more temporally-variable in nature than terrestrial coastlines. Additionally, knowledge of the underlying rock type is severely limited due to inability to access much of the geology through the ice (Stål et al., 2019). However, even in this homogeneous environment, one might expect a relatively higher complexity due to the presence of glacial valleys an example would include the western Peninsula's fjord-like coast, where there are glacial erosion processes in motion. Glacial erosive processes have a distinct signature (Anderson et al., 2006) that would result in a higher coastal complexity. Although the formative processes may differ between Antarctic and terres-

trial scenarios, the methodology does not assume prescriptive or formative processes but classifies purely on differences in complexity over a range of length scales. The analysis of Cx using this multi-scale approach also allows the identification and analysis of morphologically similar coastal environments and forms the basis for further research into their relationship to, and synergy with natural processes."

To clarify the point of 'how frequently should the resurvey take place?', we have added a paragraph in the 'Conclusion' e.g. "Given the temporally variable nature of ice and as to the question of how frequently the complexity of the Antarctic coastline should be recalculated, most major change in margins happens with ice shelf advance/retreat (i.e., calving and ice front advance). Of these processes, retreat has by far a shorter timescale. So, one could argue that a re-assessment should happen in conjunction with major calving - but such events tend to be regionally limited (e.g., the calving of the Amery Ice Shelf in 2020). Ice shelf collapse (e.g., Wilkins in 2008/09) is a little more dramatic but still geographically limited. Thereby, such re-evaluations aren't needed frequently unless there's major change. Runaway grounding line retreat leading to major coastal margins changes might be sufficient grounds for re-evaluation, but this hasn't yet happened. Significance of changes could be assess using standard change detection metrics (e.g., estimate the distribution of the current coastline features, see if the new coastline complexity falls outside of this distribution) thus justifying another evaluation."

References

Anderson, R. S., Molnar, P., and Kessler, M. A.: Features of glacial valley profiles simply explained, Journal of Geophysical Research: Earth Surface, 111, 2006. Stål, T., Reading, A. M., Halpin, J. A., and Whittaker, J. M.: A multivariate approach for mapping lithospheric domain boundaries in East Antarctica, Geophysical Research Letters, 46, 10404-10416, 2019.

---

## Author Comment (AC2) · 12 Apr 2021

General comments

Ted Scambos (Reviewer)#2 ()

'The authors have produced an interesting and first-ever (for Antarctica) analysis of coastal complexity using methods previously used on other continents (Australia, at least). They describe the mathematics and statistics of the approach quite well (it seems clearly written and reasoned, and the math is not opaque), and note the - potential- importance and usefulness of the analysis in terms of studies of coastal ice and ocean conditions, and biodiversity. The study is well-written, and I have mostly minor comments. I would encourage the authors (and editor) to consider asking for some kind of 'case study' or close-up assessment of a region of interest, perhaps near Davis

Base or McMurdo (places with lots of existing literature) and describe in qualitative or semi-quantitative terms how the coastal type is reflected in the physical characteristics of the sea ice, biological environment and variability, or geology and erosion history..... it would make a stronger case that this is not just a topological exercise, but something that could potentially be mined for scientific application. The authors do a fair job of making this case in general terms, but two or three paragraphs looking at the results in a particular area would be a good addition. Not an essential addition for publication, but one that would be nice to see.

Response: To Dr Scambos (Reviewer #2), we thank you for taking the time to read and review our manuscript. Your insightful comments and observations have not only contributed to the improvement of our manuscript but have generated new ideas that we are keen to pursue. We have revised our original submission with your input and will submit the revised manuscript, along with a version containing all the changes made. Many thanks again.

We have addressed the regional case suggestion by expanding upon Figs 10 and 11. One problem is that localised case studies quickly become "non-localised" when the multiscale complexity is fully analysed (largest scale is +/- 256 km). We could analyse a subset of these length scales to ensure a case study is localised, but this would some-what negate the point of the study - examining patterns at a meso or continental-wide scale. However, we have added more explanatory text to Figure 10 e.g., 'Lützow-Holm Bay is so deeply embayed that it shelters and favours formation of multiyear fast ice, whereas along Enderby Land coast, its much higher degree of exposure permits fast ice to form only seasonally'. Additionally, more text was added to Figure 11 e.g., 'Fast ice forms readily within the protection of McMurdo Sound, but with incrementally lower persistence further north (Fraser et al., 2020)'. The northern and southern side of the Drygalsky Ice Tongue have identical complexity classes but very different glaciol-ogy (sometimes fast ice to the south, but almost always ice-free/coastal polynya to the north. This highlights the importance of considering embayment aspect, which is

independent of complexity.'

Supplementary comments - Reviewer #2

Comment: L#15 no comma needed between 'novel technique' Response: Removed comma

Comment: L#22 suggest ending the sentence at '. . .biological studies'. The last part has a circular reasoning tone to it. L30 change to '. . .of terrestrial land areas.' Response: Ended sentence at '. . .biological studies' as suggested

Comment: L#30 change to '. . .of terrestrial land areas.' Response: Text changed to '. . .of terrestrial land areas.' as suggested

Comment: L#37 is it really more challenging to quantify? Is it not just a mathematical analysis no matter the shape, or pace of change? Is change in the coastline over time considered somehow? Response: Text changed from '. . .therefore more challenging to quantify. . .' to '. . .thereby a different challenge to terrestrial coastlines. . .'

Comment: L#40 I don't think the case for the importance of coastal complexity has really be made at this point in the text. Suggest moving this sentence to lead-off the paragraph that begins on Line70. Response: Text 'Despite its importance, no study has quantified the coastline complexity of the Antarctic continent.' moved to suggested position .

Comment: L#54 suggest '. . .and has a major impact on logistical operations, e.g., station resupply.' Response: Text changed from '. . .and has a major role in logistical operations e.g., station resupply.' to '. . .and has a major impact on logistical operations, e.g., station resupply.'

Comment: L#79 I think the text flow would be better if you moved this paragraph up one step and then combined the one above it and the one below it as the last statement before launching into the study. Response: Text (at end of Introduction) restructured as suggested,

[revised manuscript text omitted]

Comment: L#183-187 An interesting observation. I think the western-bias in West Antarctica may be an indirect function of climate and winds: much of the 0-60°W coastline (in your broad use of the word) is an ice shelf margin; but in the $60 - 180°$W region,

warmer conditions along the western Peninsula mean a more fjord-like coast, greater coastal complexity, and a strong westward orientation of both bays and peninsulas. On the Amundsen coast, there is a strong easterly prevailing wind direction, which may influence calving and ice shelf structure. It would be a nice connection to the physical world to discuss the broad trends in this way (I'm not saying that my speculation is correct, but I think there is a physical / climate link to these obs of East and West Antarctica's coast).

Response: The authors are aware of the recent Nature paper by Holland et al., (2019) which describes the accelerated melting of the western Antarctic ice sheet caused by changing wind patterns due to anthropogenic-influenced climate change. Thereby generating strong easterlies bringing warm ocean water onto the continental shelf and into contact with the West Antarctic ice sheet. It would be interesting to see how regional coastal complexity and the orientation of bays and peninsulas influence or respond to the intrusion of these warmer ocean waters on the Amundsen Sea continental shelf, but for the moment the authors feel it is presently beyond the remit of this study and would be best suited to a future project.

Comment: L#218 – change 'carve' to 'calve' Response: Text changed from 'carve' to 'calve'

Comment: L#233-234 Would it be useful / illuminating to include a third enlargement, perhaps of some part of the Peninsula? (I'm thinking southwest area or northern half on both sides). Response: We used Figure 11 to illustrate a contrasting feature e.g., 'Fast ice forms readily within the protection of McMurdo Sound, but with incrementally lower persistence further north (Fraser et al., 2020)". The northern and southern side of the Drygalski Ice Tongue have identical complexity classes but very different glaciology (sometimes fast ice to the south, but almost always ice-free/polynya to the north. This highlights the role of embayment aspect, which is independent of complexity.'

Comment: Figure 6: I can't detect a difference between the left and the right diagram

– ah, ok a very slight change in the red-blue boundary area. . .. But is this worth a separate figure? problem there? Response: Diagram reduced to a single figure.

Comment: Figure 11: change to 'Enlargement of the western Ross Sea coastal area. . .' Response: Text changed to 'Enlargement of the western Ross Sea coastal area'

---

## Author Response (AR2)

**Comments to the Topical Editor:**

We have adjusted the manuscript according to your suggestions below:

(1) Please add the DOIs to the references (as "https://doi.org/" code. Most citations are from journal publications which surely have DOIs (I usually use a simple Google search with the full reference)

(2) For Mohajer et al. (2011), please change " arXiv preprint arXiv:1103.4767, 2011. 2011" to " arXiv preprint available at https://arxiv.org/abs/1103.4767", 2011"

(3) Please make sure that all DOIs and URLs are "executable", i.e. have active links